# Self-Supervised Graph Transformer on Large-Scale Molecular Data

**Yu Rong**[1]*, **Yatao Bian**[1]*, **Tingyang Xu**[1], **Weiyang Xie**[1], **Ying Wei**[1],
**Wenbing Huang**[2]†, **Junzhou Huang**[1]
[1]Tencent AI Lab
[2] Beijing National Research Center for Information Science and Technology (BNRist),
Department of Computer Science and Technology, Tsinghua University
yu.rong@hotmail.com, yatao.bian@gmail.com, {tingyangxu, weiyangxie}@tencent.com
judyweiying@gmail.com, hwenbing@126.com, jzhuang@uta.edu

## Abstract

How to obtain informative representations of molecules is a crucial prerequisite
in AI-driven drug design and discovery. Recent researches abstract molecules as
graphs and employ Graph Neural Networks (GNNs) for molecular representation
learning. Nevertheless, two issues impede the usage of GNNs in real scenarios:
(1) insufficient labeled molecules for supervised training; (2) poor generalization
capability to new-synthesized molecules. To address them both, we propose a
novel framework, GROVER, which stands for **G**raph **R**epresentation fr**O**m self-
super**V**ised m**E**ssage passing t**R**ansformer. With carefully designed self-supervised
tasks in node-, edge- and graph-level, GROVER can learn rich structural and seman-
tic information of molecules from enormous unlabelled molecular data. Rather, to
encode such complex information, GROVER integrates Message Passing Networks
into the Transformer-style architecture to deliver a class of more expressive en-
coders of molecules. The flexibility of GROVER allows it to be trained efficiently
on large-scale molecular dataset without requiring any supervision, thus being
immunized to the two issues mentioned above. We pre-train GROVER with 100
million parameters on 10 million unlabelled molecules—the biggest GNN and the
largest training dataset in molecular representation learning. We then leverage the
pre-trained GROVER for molecular property prediction followed by task-specific
fine-tuning, where we observe a huge improvement (more than 6% on average)
from current state-of-the-art methods on 11 challenging benchmarks. The insights
we gained are that well-designed self-supervision losses and largely-expressive
pre-trained models enjoy the significant potential on performance boosting.

## 1 Introduction

Inspired by the remarkable achievements of deep learning in many scientific domains, such as com-
puter vision [57, 19], natural language processing [53, 51], and social networks [31, 3], researchers
are exploiting deep learning approaches to accelerate the process of drug discovery and reduce costs
by facilitating the rapid identification of molecules [5]. Molecules can be naturally represented by
molecular graphs which preserve rich structural information. Therefore, supervised deep learning of
graphs, especially with Graph Neural Networks(GNNs) [46, 25] have shown promising results in
many tasks, such as molecular property prediction [13, 23] and virtual screening [56, 64].

Despite the fruitful progress, two issues still impede the usage of deep learning in real scenarios: (1) insufficient labeled data for molecular tasks; (2) poor generalization capability of models in the enormous chemical space. Different from other domains (such as image classification) that have rich-source labeled data, getting labels of molecular property requires wet-lab experiments which is time-consuming and resource-costly. As a consequence, most public molecular benchmarks contain far-from-adequate labels. Conducting deep learning on these benchmarks is prone to over-fitting and the learned model can hardly cope with the out-of-distribution molecules.

Indeed, it has been a long-standing goal in deep learning to improve the generalization power of neural networks. Towards this goal, certain progress has been made. For example, in Natural Language Processing (NLP), researchers can pre-train the model from large-scale unlabeled sentences via a newly-proposed technique—the self-supervised learning. Several successful self-supervised pretraining strategies, such as BERT [9] and GPT [38] have been developed to tackle a variety of language tasks. By contending that molecule can be transformed into sequential representation— SMILES [59], the work by [58] tries to adopt the BERT-style method to pretrain the model, and Liu et.al. [29] also exploit the idea from N-gram approach in NLP and conducts vertices embedding by predicting the vertices attributes. Unfortunately, these approaches fail to explicitly encode the structural information of molecules as using the SMILES representation is not topology-aware.

Without using SMILES, several works aim to establish a pre-trained model directly on the graph representations of molecules. Hu et.al. [18] investigate the strategies to construct the three pre-training tasks, i.e., context prediction and node masking for node-level self-supervised learning and graph property prediction for graph-level pre-training. We argue that the formulation of pre-training in this way is suboptimal. First, in the masking task, they treat the atom type as the label. Different from NLP tasks, the number of atom types in molecules is much smaller than the size of a language vocabulary. Therefore, it would suffer from serious representation ambiguity and the model is hard to encode meaningful information especially for the highly frequent atoms. Second, the graph-level pre-training task in [18] is supervised. This limits the usage in practice since most of molecules are completely unlabelled, and it also introduces the risk of *negative transfer* for the downstream tasks if they are inconsistent to the graph-level supervised loss.

In this paper, we improve the pre-training model for molecular graph by introducing a novel molecular representation framework, GROVER, namely, **G**raph **R**epresentation fr**O**m self-super**V**ised m**E**ssage passing t**R**ansformer. GROVER constructs two types of self-supervised tasks. For the node/edge-level tasks, instead of predicting the node/edge type alone, GROVER randomly masks a local subgraph of the target node/edge and predicts this contextual property from node embeddings. In this way, GROVER can alleviate the ambiguity problem by considering both the target node/edge and its context being masked. For the graph-level tasks, by incorporating the domain knowledge, GROVER extracts the semantic motifs existing in molecular graphs and predicts the occurrence of these motifs for a molecule from graph embeddings. Since the semantic motifs can be obtained by a low-cost pattern matching method, GROVER can make use of any molecular to optimize the graph-level embedding. With self-supervised tasks in node-, edge- and graph-level, GROVER can learn rich structural and semantic information of molecules from enormous unlabelled molecular data. Rather, to encode such complex information, GROVER integrates Message Passing Networks with the Transformer-style architecture to deliver a class of highly expressive encoders of molecules. The flexibility of GROVER allows it to be trained efficiently on large-scale molecular data without requiring any supervision. We pre-train GROVER with 100 million parameters on 10 million of unlabelled molecules—the biggest GNN and the largest training dataset that have been applied. We then leverage the pre-trained GROVER models to downstream molecular property prediction tasks followed by task-specific fine-tuning. On the downstream tasks, GROVER achieve 22.4% relative improvement compared with [29] and 7.4% relative improvement compared with [18] on classification tasks. Furthermore, even compared with current state-of-the-art results for each data set, we observe a huge relative improvement of GROVER (more than 6% on average) over 11 popular benchmarks.

## 2 Related Work

**Molecular Representation Learning.** To represent molecules in the vector space, the traditional chemical fingerprints, such as ECFP [42], try to encode the neighbors of atoms in the molecule into a fix-length vector. To improve the expressive power of chemical fingerprints, some studies [10, 7] introduce convolutional layers to learn the neural fingerprints of molecules, and apply the neural

fingerprints to the downstream tasks, such as property prediction. Following these works, [62, 21] take the SMILES representation [59] as input and use RNN-based models to produce molecular representations. Recently, many works [23, 46, 45] explore the graph convolutional network to encode molecular graphs into neural fingerprints. A slot of work [44, 61] propose to learn the aggregation weights by extending the Graph Attention Network (GAT) [54]. To better capture the interactions among atoms, [13] proposes to use a message passing framework and [63, 25] extend this framework to model bond interactions. Furthermore, [30] builds a hierarchical GNN to capture multilevel interactions.

**Self-supervised Learning on Graphs.** Self-supervised learning has a long history in machine learning and has achieved fruitful progresses in many areas, such as computer vision [35] and language modeling [9]. The traditional graph embedding methods [37, 14] define different kinds of graph proximity, i.e., the vertex proximity relationship, as the self-supervised objective to learn vertex embeddings. GraphSAGE [15] proposes to use a random-walk based proximity objective to train GNN in an unsupervised fashion. [55, 36, 50] exploit the mutual information maximization scheme to construct objective for GNNs. Recently, two works are proposed to construct unsupervised representations for molecular graphs. Liu et.al. [29] employ an N-gram model to extract the context of vertices and construct the graph representation by assembling the vertex embeddings in short walks in the graph. Hu et.al. [18] investigate various strategies to pre-train the GNNs and propose three self-supervised tasks to learn molecular representations. However, [18] isolates the highly correlated tasks of context prediction and node/edge type prediction, which makes it difficult to preserve domain knowledge between the local structure and the node attributes. Besides, the graph-level task in [18] is constructed by the supervised property labels, which is impeded by the limited number of supervised labels of molecules and has demonstrated the negative transfer in the downstream tasks. Contrast with [18], the molecular representations derived by our method are more appropriate in terms of persevering the domain knowledge, which has demonstrated remarkable effectiveness in downstream tasks without negative transfer.

## 3    Preliminaries of Transformer-style Models and Graph Neural Networks

We briefly introduce the concepts of supervised graph learning, Transformer [53], and GNNs in this section.

**Supervised learning tasks of graphs.** A molecule can be abstracted as a graph $G = (\mathcal{V}, \mathcal{E})$, where $|\mathcal{V}| = n$ refers to a set of $n$ nodes (atoms) and $|\mathcal{E}| = m$ refers to a set of $m$ edges (bonds) in the molecule. $\mathcal{N}_v$ is used to denote the set of neighbors of node $v$. We use $\mathbf{x}_v$ to represent the initial features of node $v$, and $\mathbf{e}_{uv}$ as the initial features of edge $(u, v)$. In graph learning, there are usually two categories of supervised tasks: i) *Node classification/regression*, where each node $v$ has a label/target $y_v$, and the task is to learn to predict the labels of unseen nodes; ii) *Graph classification/regression*, where a set of graphs $\{G_1, ..., G_N\}$ and their labels/targets $\{y_1, ..., y_N\}$ are given, and the task is to predict the label/target of a new graph.

**Attention mechanism and the Transformer-style architectures.** The attention mechanism is the main building block of Transformer. We focus on multi-head attention, which stacks several scaled dot-product attention layers together and allows parallel running. One scaled dot-product attention layer takes a set of queries, keys, values $(\mathbf{q}, \mathbf{k}, \mathbf{v})$ as inputs. Then it computes the dot products of the query with all keys, and applies a softmax function to obtain the weights on the values. By stacking the set of $(\mathbf{q}, \mathbf{k}, \mathbf{v})$s into matrices $(\mathbf{Q}, \mathbf{K}, \mathbf{V})$, it admits highly optimized matrix multiplication operations. Specifically, the outputs can be arranged as a matrix:

$$\text{Attention}(\mathbf{Q}, \mathbf{K}, \mathbf{V}) = \text{softmax}(\mathbf{Q}\mathbf{K}^{\top}/\sqrt{d})\mathbf{V}, \tag{1}$$

where $d$ is the dimension of $\mathbf{q}$ and $\mathbf{k}$. Suppose we arrange $k$ attention layers into the multi-head attention, then its output matrix can be written as,

$$\text{MultiHead}(\mathbf{Q}, \mathbf{K}, \mathbf{V}) = \text{Concat}(\text{head}_1, ..., \text{head}_k)\mathbf{W}^O,$$
$$\text{head}_i = \text{Attention}(\mathbf{Q}\mathbf{W}_i^{\mathbf{Q}}, \mathbf{K}\mathbf{W}_i^{\mathbf{K}}, \mathbf{V}\mathbf{W}_i^{\mathbf{V}}), \tag{2}$$

where $\mathbf{W}_i^{\mathbf{Q}}, \mathbf{W}_i^{\mathbf{K}}, \mathbf{W}_i^{\mathbf{V}}$ are the projection matrices of head $i$.

**Graph Neural Networks (GNNs).** Recently, GNNs have received a surge of interest in various domains, such as knowledge graph, social networks and drug discovery. The key operation of

GNNs lies in a message passing process, which involves message passing (also called neighborhood aggregation) between the nodes in the graph. The message passing operation iteratively updates a node $v$'s hidden states, $\mathbf{h}_v$, by aggregating the hidden states of $v$'s neighboring nodes and edges. In general, the message passing process involves several iterations, each iteration can be further partitioned into several hops. Suppose there are $L$ iterations, and iteration $l$ contains $K_l$ hops. Formally, in iteration $l$, the $k$-th hop can be formulated as,

$$\mathbf{m}_v^{(l,k)} = \text{AGGREGATE}^{(l)}(\{(\mathbf{h}_v^{(l,k-1)}, \mathbf{h}_u^{(l,k-1)}, \mathbf{e}_{uv}) \mid u \in \mathcal{N}_v\}), \tag{3}$$
$$\mathbf{h}_v^{(l,k)} = \sigma(\mathbf{W}^{(l)}\mathbf{m}_v^{(l,k)} + \mathbf{b}^{(l)}),$$

where $\mathbf{m}_v^{(l,k)}$ is the aggregated message, and $\sigma(\cdot)$ is some activation function. We make the convention that $\mathbf{h}_v^{(l,0)} := \mathbf{h}_v^{(l-1,K_{l-1})}$. There are several popular ways of choosing $\text{AGGREGATE}^{(l)}(\cdot)$, such as mean, max pooling and graph attention mechanism [54, 15]. For one iteration of message passing, there are a layer of trainable parameters (i.e., parameters inside $\text{AGGREGATE}^{(l)}(\cdot)$, $\mathbf{W}^{(l)}$ and $\mathbf{b}^{(l)}$. These parameters are shared across the $K_l$ hops within the iteration $l$. After $L$ iterations of message passing, the hidden states of the last hop in the last iteration are used as the embeddings of the nodes, i.e., $\mathbf{h}_v^{(L,K_L)}, v \in \mathcal{V}$. Lastly, a READOUT operation is applied to get the graph-level representation,

$$\mathbf{h}_G = \text{READOUT}(\{\mathbf{h}_v^{(0,K_0)}, ..., \mathbf{h}_v^{(L,K_L)} \mid v \in \mathcal{V}\}). \tag{4}$$

## 4  The GROVER Pre-training Framework

This section contains details of our pre-training architecture together with the well-designed self-supervision tasks. On a high level, the model is a Transformer-based neural network with tailored GNNs as the self-attention building blocks. The GNNs therein enable capturing structural information in the graph data and information flow on both the node and edge message passing paths. Furthermore, we introduce a dynamic message passing scheme in the tailored GNN, which is proved to boost the generalization performance of GROVER models.

### 4.1  Details of Model Architecture

GROVER consists of two modules: the node GNN transformer and edge GNN transformer. In order to ease the exposition, we will only explain details of the node GNN transformer (abbreviated as node GTransformer) in the sequel, and ignore the edge GNN transformer since it has a similar structure. Figure 1 demonstrates the overall architecture of node GTransformer. More details of GROVER are deferred to Appendix A.

**GNN Transformer** (GTransformer). The key component of the node GTransformer is our proposed *graph multi-head attention* component, which is the attention blocks tailored to structural input data. A vanilla attention block, such as that in Equation (1), requires vectorized inputs. However, graph inputs are naturally structural data that are not vectorized. So we design a tailored GNNs (dyMPN, see the following sections for details) to extract vectors as queries, keys and values from nodes of the graph, then feed them into the attention block.

This strategy is *simple* yet *powerful*, because it enables utilizing the highly expressive GNN models, to better model the structural information in molecular data. The high expressiveness of GTransformer can be attributed to its *bi-level* information extraction framework. It is well-known that the message passing process captures local structural information of the graph, therefore using the outputs of the GNN model as queries, keys and values

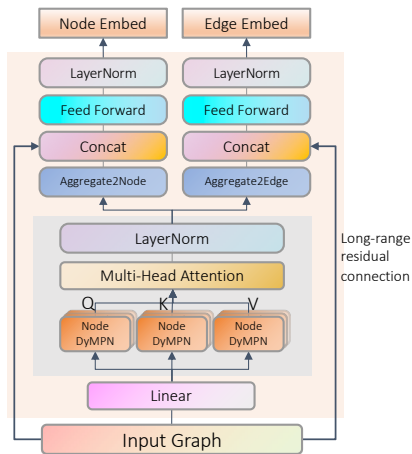

Figure 1: Overview of GTransformer.

would get the local subgraph structure involved, thus constituting the first level of information extraction. Meanwhile, the Transformer encoder can be viewed as a variant of the GAT [54, 22] on a fully connected graph constructed by $\mathcal{V}$. Hence, using Transformer encoder on top of these queries,

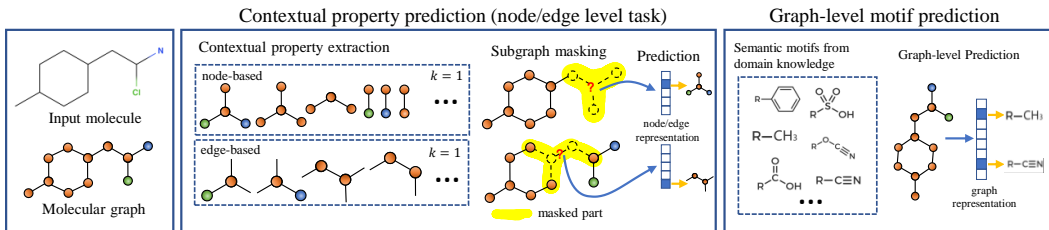

Figure 2: Overview of the designed self-supervised tasks of GROVER.

keys and values makes it possible to extract global relations between nodes, which enables the second level of information extraction. This bi-level information extraction strategy largely enhances the representational power of GROVER models.

Additionally, GTransformer applies a *single long-range residual connection* from the input feature to convey the initial node/edge feature information directly to the last layers of GTransformer instead of multiple short-range residual connections in the original Transformer architecture. Two benefits could be obtained from this single long-range residual connection: i) like ordinary residual connections, it improves the training process by alleviating the vanishing gradient problem [17], ii) compared to the various short-range residual connections in the Transformer encoder, our long-range residual connection can alleviate the *over-smoothing* [34, 20] problem in the message passing process.

**Dynamic Message Passing Network** (dyMPN). The general message passing process (see Equation (3)) has two hyperparameters: number of iterations/layers $L$ and number of hops $K_l, l = 1, ..., L$ within each iteration. The number of hops is closely related to the size of the receptive field of the graph convolution operation, which would affect generalizability of the message passing model.

Given a fixed number of layers $L$, we find out that the pre-specified number of hops might not work well for different kinds of dataset. Instead of pre-specified $K_l$, we develop a randomized strategy for choosing the number of message passing hops during training process: at each epoch, we choose $K_l$ from some random distribution for layer $l$. Two choices of randomization work well: i) $K_l \sim U(a, b)$, drawn from a uniform distribution; ii) $K_l$ is drawn from a truncated normal distribution $\phi(\mu, \sigma, a, b)$, which is derived from that of a normally distributed random variable by bounding the random variable from both bellow and above. Specifically, let its support be $x \in [a, b]$, then the p.d.f. is $f(x) = \frac{\frac{1}{\sqrt{2\pi}} \exp\left[-\frac{1}{2}(\frac{x-\mu}{\sigma})^2\right]}{\sigma[\Phi(\frac{b-\mu}{\sigma}) - \Phi(\frac{a-\mu}{\sigma})]}$, where $\Phi(x) = \frac{1}{2}(1 + \text{erf}(\frac{x}{\sqrt{2}}))$ is the cumulative distribution of a standard normal distribution.

The above randomized message passing scheme enables random receptive field for each node in graph convolution operation. We call the induced network Dynamic Message Passing networks (abbreviated as dyMPN). Extensive experimental verification demonstrates that dyMPN enjoys better generalization performance than vanilla message passing networks without the randomization strategy.

## 4.2 Self-supervised Task Construction for Pre-training

The success of the pre-training model crucially depends on the design of self-supervision tasks. Different from Hu et.al. [18], to avoid negative transfer on downstream tasks, we do not use the supervised labels in pre-training and propose new self-supervision tasks on both of these two levels: *contextual property prediction* and *graph-level motif prediction*, which are sketched in Figure 2.

**Contextual Property Prediction.** A good self-supervision task on the node level should satisfy the following properties: 1) The prediction target is reliable and easy to get; 2) The prediction target should reflect contextual information of the node/edge. Guided by these criteria, we present the tasks on both nodes and edges. They both try to predict the context-aware properties of the target node/edge within some local subgraph. What kinds of context-aware properties shall one use? We define recurrent statistical properties of local subgraph in the following two-step manner (let us take the node subgraph

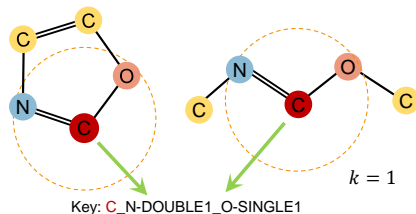

Figure 3: Illustration of contextual properties.

in Figure 3 as the example): i) Given a target node (e.g., the Carbon atom in red color), we extract its local subgraph as its $k$-hop neighboring nodes and edges. When $k$=1, it involves the Nitrogen atom, Oxygen atom, the double bond and single bond. ii) We extract statistical properties of this subgraph, specifically, we count the number of occurrence of (node, edge) pairs around the center node, which makes the term of `node-edge-counts`. Then we list all the node-edge counts terms in alphabetical order, which constitutes the final property: e.g., `C_N-DOUBLE1_O-SINGLE1` in the example. This step can be viewed as a clustering process: the subgraphs are clustered according to the extracted properties, one property corresponds to a cluster of subgraphs with the same statistical property.

With the context-aware property defined, the contextual property prediction task works as follows: given a molecular graph, after feeding it into the GROVER encoder, we obtain embeddings of its atoms and bonds. Suppose randomly choose the atom $v$ and its embedding is $\mathbf{h}_v$. Instead of predicting the atom type of $v$, we would like $\mathbf{h}_v$ to encode some contextual information around node $v$. The way to achieve this target is to feed $\mathbf{h}_v$ into a *very* simple model (such as a fully connected layer), then use the output to predict the contextual properties of node $v$. This prediction is a multi-class prediction problem (one class corresponds to one contextual property).

**Graph-level Motif Prediction.** Graph-level self-supervision task also needs reliable and cheap labels. Motifs are recurrent sub-graphs among the input graph data, which are prevalent in molecular graph data. One important class of motifs in molecules are functional groups, which encodes the rich domain knowledge of molecules and can be easily detected by the professional software, such as RDKit [27]. Formally, the motif prediction task can be formulated as a *multi-label* classification problem, where each motif corresponds to one label. Suppose we are considering the presence of $p$ motifs $\{\mathbf{m}_1, ..., \mathbf{m}_p\}$ in the molecular data. For one specific molecule (abstracted as a graph $G$), we use RDKit to detect whether each of the motif shows up in $G$, then use it as the target of the motif prediction task.

## 4.3 Fine-tuning for Downstream Tasks

After pre-training GROVER models on massive unlabelled data with the designed self-supervised tasks, one should obtain a high-quality molecular encoder which is able to output embeddings for both nodes and edges. These embeddings can be used for downstream tasks through the fine-tuning process. Various downstream tasks could benefit from the pre-trained GROVER models. They can be roughly divided into three categories: node level tasks, e.g., node classification; edge level tasks, e.g., link prediction; and graph level tasks, such as the property prediction for molecules. Take the graph level task for instance. Given node/edge embeddings output by the GROVER encoder, we can apply some READOUT function (Equation (4)) to get the graph embedding firstly, then use additional multiple layer perceptron (MLP) to predict the property of the molecular graph. One would use part of the supervised data to fine-tune both the encoder and additional parameters (READOUT and MLP). After several epochs of fine-tuning, one can expect a well-performed model for property prediction.

## 5 Experiments

**Pre-training Data Collection.** We collect 11 million (M) unlabelled molecules sampled from ZINC15 [48] and Chembl [11] datasets to pre-train GROVER. We randomly split 10% of unlabelled molecules as the validation sets for model selection.

**Fine-tuning Tasks and Datasets.** To thoroughly evaluate GROVER on downstream tasks, we conduct experiments on 11 benchmark datasets from the MoleculeNet [60] with various targets, such as quantum mechanics, physical chemistry, biophysics and physiology.[3] Details are deferred to Appendix B.1. In machine learning tasks, random splitting is a common process to split the dataset. However, for molecular property prediction, scaffold splitting [2] offers a more challenging yet realistic way of splitting. We adopt the scaffold splitting method with a ratio for train/validation/test as 8:1:1. For each dataset, as suggested by [60], we apply three independent runs on three random-seeded scaffold splitting and report the mean and standard deviations.

**Baselines.** We comprehensively evaluate GROVER against 10 popular baselines from MoleculeNet [60] and several state-of-the-arts (STOAs) approaches. Among them, TF_Roubust [40] is a DNN-based mulitask framework taking the molecular fingerprints as the input. GraphConv [24], Weave

Table 1: The performance comparison. The numbers in brackets are the standard deviation. The methods in green are pre-trained methods.

| Classification (Higher is better) | | | | | | |
|---|---|---|---|---|---|---|
| Dataset | BBBP | SIDER | ClinTox | BACE | Tox21 | ToxCast |
| # Molecules | 2039 | 1427 | 1478 | 1513 | 7831 | 8575 |
| TF_Robust [40] | $0.860_{(0.087)}$ | $0.607_{(0.033)}$ | $0.765_{(0.085)}$ | $0.824_{(0.022)}$ | $0.698_{(0.012)}$ | $0.585_{(0.031)}$ |
| GraphConv [24] | $0.877_{(0.036)}$ | $0.593_{(0.035)}$ | $0.845_{(0.051)}$ | $0.854_{(0.011)}$ | $0.772_{(0.041)}$ | $0.650_{(0.025)}$ |
| Weave [23] | $0.837_{(0.065)}$ | $0.543_{(0.034)}$ | $0.823_{(0.023)}$ | $0.791_{(0.008)}$ | $0.741_{(0.044)}$ | $0.678_{(0.024)}$ |
| SchNet [45] | $0.847_{(0.024)}$ | $0.545_{(0.038)}$ | $0.717_{(0.042)}$ | $0.750_{(0.033)}$ | $0.767_{(0.025)}$ | $0.679_{(0.021)}$ |
| MPNN [13] | $0.913_{(0.041)}$ | $0.595_{(0.030)}$ | $0.879_{(0.054)}$ | $0.815_{(0.044)}$ | $0.808_{(0.024)}$ | $0.691_{(0.013)}$ |
| DMPNN [63] | $0.919_{(0.030)}$ | $0.632_{(0.023)}$ | $0.897_{(0.040)}$ | $0.852_{(0.053)}$ | $0.826_{(0.023)}$ | $0.718_{(0.011)}$ |
| MGCN [30] | $0.850_{(0.064)}$ | $0.552_{(0.018)}$ | $0.634_{(0.042)}$ | $0.734_{(0.030)}$ | $0.707_{(0.016)}$ | $0.663_{(0.009)}$ |
| AttentiveFP [61] | $0.908_{(0.050)}$ | $0.605_{(0.060)}$ | $0.933_{(0.020)}$ | $0.863_{(0.015)}$ | $0.807_{(0.020)}$ | $0.579_{(0.001)}$ |
| N-GRAM [29] | $0.912_{(0.013)}$ | $0.632_{(0.005)}$ | $0.855_{(0.037)}$ | $0.876_{(0.035)}$ | $0.769_{(0.027)}$ | -[4] |
| HU. et.al[18] | $0.915_{(0.040)}$ | $0.614_{(0.006)}$ | $0.762_{(0.058)}$ | $0.851_{(0.027)}$ | $0.811_{(0.015)}$ | $0.714_{(0.019)}$ |
| GROVER$_{base}$ | $0.936_{(0.008)}$ | $0.656_{(0.006)}$ | $0.925_{(0.013)}$ | $0.878_{(0.016)}$ | $0.819_{(0.020)}$ | $0.723_{(0.010)}$ |
| GROVER$_{large}$ | $\mathbf{0.940}_{(0.019)}$ | $\mathbf{0.658}_{(0.023)}$ | $\mathbf{0.944}_{(0.021)}$ | $\mathbf{0.894}_{(0.028)}$ | $\mathbf{0.831}_{(0.025)}$ | $\mathbf{0.737}_{(0.010)}$ |

| Regression (Lower is better) | | | | | |
|---|---|---|---|---|---|
| Dataset | FreeSolv | ESOL | Lipo | QM7 | QM8 |
| # Molecules | 642 | 1128 | 4200 | 6830 | 21786 |
| TF_Robust [40] | $4.122_{(0.085)}$ | $1.722_{(0.038)}$ | $0.909_{(0.060)}$ | $120.6_{(9.6)}$ | $0.024_{(0.001)}$ |
| GraphConv [24] | $2.900_{(0.135)}$ | $1.068_{(0.050)}$ | $0.712_{(0.049)}$ | $118.9_{(20.2)}$ | $0.021_{(0.001)}$ |
| Weave [23] | $2.398_{(0.250)}$ | $1.158_{(0.055)}$ | $0.813_{(0.042)}$ | $94.7_{(2.7)}$ | $0.022_{(0.001)}$ |
| SchNet [45] | $3.215_{(0.755)}$ | $1.045_{(0.064)}$ | $0.909_{(0.098)}$ | $74.2_{(6.0)}$ | $0.020_{(0.002)}$ |
| MPNN [13] | $2.185_{(0.952)}$ | $1.167_{(0.430)}$ | $0.672_{(0.051)}$ | $113.0_{(17.2)}$ | $0.015_{(0.002)}$ |
| DMPNN [63] | $2.177_{(0.914)}$ | $0.980_{(0.258)}$ | $0.653_{(0.046)}$ | $105.8_{(13.2)}$ | $0.0143_{(0.002)}$ |
| MGCN [30] | $3.349_{(0.097)}$ | $1.266_{(0.147)}$ | $1.113_{(0.041)}$ | $77.6_{(4.7)}$ | $0.022_{(0.002)}$ |
| AttentiveFP [61] | $2.030_{(0.420)}$ | $0.853_{(0.060)}$ | $0.650_{(0.030)}$ | $126.7_{(4.0)}$ | $0.0282_{(0.001)}$ |
| N-GRAM [29] | $2.512_{(0.190)}$ | $1.100_{(0.160)}$ | $0.876_{(0.033)}$ | $125.6_{(1.5)}$ | $0.0320_{(0.003)}$ |
| GROVER$_{base}$ | $1.592_{(0.072)}$ | $0.888_{(0.116)}$ | $0.563_{(0.030)}$ | $\mathbf{72.5}_{(5.9)}$ | $0.0172_{(0.002)}$ |
| GROVER$_{large}$ | $\mathbf{1.544}_{(0.397)}$ | $\mathbf{0.831}_{(0.120)}$ | $\mathbf{0.560}_{(0.035)}$ | $72.6_{(3.8)}$ | $\mathbf{0.0125}_{(0.002)}$ |

[23] and SchNet [45] are three graph convolutional models. MPNN [13] and its variants DMPNN [63] and MGCN [30] are models considering the edge features during message passing. AttentiveFP [61] is an extension of the graph attention network. Specifically, to demonstrate the power of our self-supervised strategy, we also compare GROVER with two pre-trained models: N-Gram [29] and Hu et.al [18]. We only report classification results for [18] since the original implementation do not admit regression task without non-trivial modifications.

**Experimental Configurations.** We use Adam optimizer for both pre-train and fine-tuning. The Noam learning rate scheduler [9] is adopted to adjust the learning rate during training. Specific configurations are:

GROVER **Pre-training.** For the contextual property prediction task, we set the context radius $k = 1$ to extract the contextual property dictionary, and obtain 2518 and 2686 distinct node and edge contextual properties as the node and edge label, respectively. For each molecular graph, we randomly mask 15% of node and edge labels for prediction. For the graph-level motif prediction task, we use RDKit [27] to extract 85 functional groups as the motifs of molecules. We represent the label of motifs as the one-hot vector. To evaluate the effect of model size, we pre-train two GROVER models, GROVER$_{base}$ and GROVER$_{large}$ with different hidden sizes, while keeping all other hyper-parameters the same. Specifically, GROVER$_{base}$ contains ~48M parameters and GROVER$_{large}$ contains ~100M parameters. We use 250 Nvidia V100 GPUs to pre-train GROVER$_{base}$ and GROVER$_{large}$. Pre-training GROVER$_{base}$ and GROVER$_{large}$ took 2.5 days and 4 days respectively. For the models depicted in Section 5.2, we use 32 Nvidia V100 GPUs to pre-train the GROVER model and its variants.

**Fine-tuning Procedure.** We use the validation loss to select the best model. For each training process, we train models for 100 epochs. For hyper-parameters, we perform the random search on the validation set for each dataset and report the best results. More pre-training and fine-tuning details are deferred to Appendix C and Appendix D.

## 5.1 Results on Downstream Tasks

Table 1 documents the overall results of all models on all datasets, where the cells in gray indicate the previous SOTAs, and the cells in blue indicates the best result achieved by GROVER. Table 1 offers the following observations: (1) GROVER models consistently achieve the best performance on all datasets with large margin on most of them. The overall relative improvement is 6.1% on all datasets (2.2% on classification tasks and 10.8% on regression tasks).[5]. This remarkable boosting validates the effectiveness of the pre-training model GROVER for molecular property prediction tasks. (2) Specifically, GROVER$_{base}$ outperforms the STOAs on 8/11 datasets, while GROVER$_{large}$ surpasses the STOAs on all datasets. This improvement can be attributed to the high expressive power of the large model, which can encode more information from the self-supervised tasks. (3) In the small dataset FreeSolv with only 642 labeled molecules, GROVER gains a 23.9% relative improvement over existing SOTAs. This confirms the strength of GROVER since it can significantly help with the tasks with very little label information.

## 5.2 Ablation Studies on Design Choices of the GROVER Framework

### 5.2.1 How Useful is the Self-supervised Pre-training?

To investigate the contribution of the self-supervision strategies, we compare the performances of pre-trained GROVER and GROVER without pre-training on classification datasets, both of which follow the same hyper-parameter setting. We report the comparison of classification task in Table 2, it is not supervising that the performance of GROVER becomes worse without pre-training. The self-supervised pre-training leads to a performance boost with an average AUC increase of 3.8% over the model without pre-training. This confirms that the self-supervised pre-training strategy can learn the implicit domain knowledge and enhance the prediction performance of downstream tasks. Notably, the datasets with fewer samples, such as SIDER, ClinTox and BACE gain a larger improvement through the self-supervised pre-training. It re-confirms the effectiveness of the self-supervised pre-training for the task with insufficient labeled molecules.

### 5.2.2 How Powerful is GTransformer Backbone?

To verify the expressive power of GTransformer, we implement GIN and MPNN based on our framework. We use a toy data set with 600K unlabelled molecules to pretrain GROVER with different backbones under the same training setting with nearly the same number of parameters (38M parameters). As

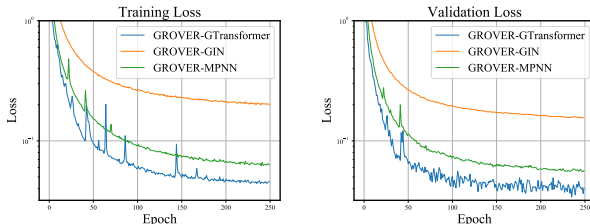

Figure 4: The training and validation losses on different backbones.

shown in Figure 4, GROVER with GTransformer backbone outperforms GIN and MPNN in both training and validation, which again verifies the effectiveness of GTransformer.

### 5.2.3 Effect of the Proposed dyMPN and GTransformer.

To justify the rationale behind the proposed GTransformer and dyMPN, we implement two variants: GROVER w/o dyMPN and GROVER w/o GTrans. GROVER w/o dyMPN fix the number of message passing hops $K_l$, while GROVER w/o GTrans replace the GTransformer with the original Transformer. We use the same toy data set to train GROVER w/o dyMPN and GROVER w/o GTrans under the same settings in Section 5.2.2. Figure 5 displays the curve of training and validation loss for three models. First, GROVER w/o GTrans is the worst one in both training and validation. It implies that trivially combining the GNN and Transformer can not enhance the expressive power of GNN. Second, dyMPN slightly harm the training loss by introducing randomness in the training process. However, the validation loss becomes better. Therefore, dyMPN brings a better generalization ability to GROVER by randomizing the receptive field for every message passing step. Overall, with new Transformer-style architecture and the dynamic message passing mechanism, GROVER enjoys high

expressive power and can well capture the structural information in molecules, thus helping with various downstream molecular prediction tasks.

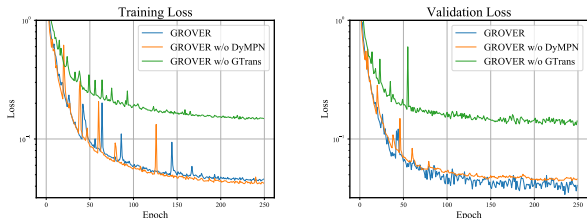

| | GROVER | No Pretrain | Abs. Imp. |
|---|---|---|---|
| BBBP (2039) | **0.940** | 0.911 | +0.029 |
| SIDER (1427) | **0.658** | 0.624 | +0.034 |
| ClinTox (1478) | **0.944** | 0.884 | +0.060 |
| BACE (1513) | **0.894** | 0.858 | +0.036 |
| Tox21 (7831) | **0.831** | 0.803 | +0.028 |
| ToxCast (8575) | **0.737** | 0.721 | +0.016 |
| Average | **0.834** | 0.803 | +0.038 |

Table 2: Comparison between GROVER with and without pre-training.

Figure 5: The training and validation loss of GROVER and its variants.

# 6   Conclusion and Future Works

We explore the potential of the large-scale pre-trained GNN models in this work. With well-designed self-supervised tasks and largely-expressive architecture, our model GROVER can learn rich implicit information from the enormous unlabelled graphs. More importantly, by fine-tuning on GROVER, we achieve huge improvements (more than $6\%$ on average) over current STOAs on 11 challenging molecular property prediction benchmarks, which first verifies the power of self-supervised pre-trained approaches in the graph learning area.

Despite the successes, there is still room to improve GNN pre-training in the following aspects: **More self-supervised tasks.** Well designed self-supervision tasks are the key of success for GNN pre-training. Except for the tasks presented in this paper, other meaningful tasks would also boost the pre-training performance, such as distance-preserving tasks and tasks that getting 3D input information involved. **More downstream tasks.** It is desirable to explore a larger category of downstream tasks, such as node prediction and link prediction tasks on different kinds of graphs. Different categories of downstream tasks might prefer different pre-training strategies/self-supervision tasks, which is worthwhile to study in the future. **Wider and deeper models.** Larger models are capable of capturing richer semantic information for more complicated tasks, as verified by several studies in the NLP area. It is also interesting to employ even larger models and data than GROVER. However, one might need to alleviate potential problems when training super large models of GNN, such as gradient vanishing and oversmoothing.

## Broader Impact

In this paper, we have developed a self-supervised pre-trained GNN model—GROVER to extract the useful implicit information from massive unlabelled molecules and the downstream tasks can largely benefit from this pre-trained GNN models. Below is the broader impact of our research:

- **For machine learning community:** This work demonstrates the success of pre-training approach on Graph Neural Networks. It is expected that our research will open up a new venue on an in-depth exploration of pre-trained GNNs for broader potential applications, such as social networks and knowledge graphs.

- **For the drug discovery community:** Researchers from drug discovery can benefit from GROVER from two aspects. First, GROVER has encoded rich structural information of molecules through the designing of self-supervision tasks. It can also produce feature vectors of atoms and molecule fingerprints, which can directly serve as inputs of downstream tasks. Second, GROVER is designed based on Graph Neural Networks and all the parameters are fully differentiable. So it is easy to fine-tune GROVER in conjunction with specific drug discovery tasks, in order to achieve better performance. We hope that GROVER can help with boosting the performance of various drug discovery applications, such as molecular property prediction and virtual screening.

## Acknowledgements and Disclosure of Funding

This work is jointly supported by Tencent AI Lab Rhino-Bird Visiting Scholars Program (VS202006), China Postdoctoral Science Foundation (Grant No.2020M670337), and the National Natural Science Foundation of China (Grant No. 62006137). The GPU resources and distributed training optimization are supported by Tencent Jizhi Team. We would thank the anonymous reviewers for their valuable suggestions. Particularly, Yu Rong wants to thank his wife, Yunman Huang, for accepting his proposal for her hand in marriage.

## Footnotes

†Wenbing Huang is the corresponding author.

[3]All datasets can be downloaded from `http://moleculenet.ai/datasets-1`

[4]The result is not presented since N-Gram on ToxCast is too time consuming to be finished in time.

[5]We use relative improvement [52] to provide the unified descriptions.

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
