[Supplementary Material]

# Appendix

## A  The Overall Architecture of GROVER Model

Figure 6: Overview of the whole GROVER architecture with both node-view GTransformer (in pink background) and edge-view GTransformer (in green background)

Figure 6 illustrates the complete architecture of GROVER models, which contains a node-view GTransformer (in pink background) and an edge-view GTransformer (in green background). Brief presentations of the node-view GTransformer have been introduced in the main text, and the edge-view GTransformer is in a similar structure. Here we elaborate more details of the GROVER model and its associated four sets of output embeddings.

As shown in Figure 6, node-view GTransformer contains node dyMPN, which maintains hidden states of nodes $\mathbf{h}_v, v \in \mathcal{V}$ and performs the message passing over nodes. Meanwhile, edge-view GTransformer contains edge dyMPN, that maintains hidden states of edges $\mathbf{h}_{vw}, \mathbf{h}_{wv}, (v, w) \in \mathcal{E}$ and conducts message passing over *edges*. The edge message passing is viewed as an ordinary message passing over the line graph of the original graph, where the line graph describes the neighboring of edges in the original graph and enables an appropriate way to define message passing over edges [6]. Note that edge hidden states have directions, i.e., $\mathbf{h}_{vw}$ is not identical to $\mathbf{h}_{wv}$ in general.

Then, after the multi-head attention, we denote the transformed node and edge hidden states by $\bar{\mathbf{h}}_v$ and $\bar{\mathbf{h}}_{vw}$, respectively.

Given the above setup, we can explain why GROVER will output four sets of embeddings in Figure 6. Let us focus on the information flow in the pink panel of Figure 6, first. Here the node hidden states $\bar{\mathbf{h}}_v$ encounter the two components, Aggregate2Node and Aggregate2Edge, which are used to aggregate the node hidden states to node messages and edge messages, respectively. Specifically, the Aggregate2Node and Aggregate2Edge components in node-view GTransformer is formulated as follows:

$$\mathbf{m}_v^{\text{node-embedding-from-node-states}} = \sum_{u \in \mathcal{N}_v} \bar{\mathbf{h}}_u \tag{5}$$

$$\mathbf{m}_{vw}^{\text{edge-embedding-from-node-states}} = \sum_{u \in \mathcal{N}_v \setminus w} \bar{\mathbf{h}}_u. \tag{6}$$

Then the node-view GTransformer transforms the node messages $\mathbf{m}_v^{\text{node-embedding-from-node-states}}$ and edge messages $\mathbf{m}_{vw}^{\text{edge-embedding-from-node-states}}$ through Pointwise Feed Forward layers [53] and Add&LayerNorm to produce the final node embeddings and edge embeddings, respectively.

Figure 7: Examples of constructing contextual properties for edges

Similarly, for the information flow in the green panel, the edge hidden states $\bar{\mathbf{h}}_{vw}$ encounter the two components Aggregate2Node and Aggregate2Edge as well. Their operations are formulated as follows,

$$\mathbf{m}_v^{\text{node-embedding-from-edge-states}} = \sum_{u \in \mathcal{N}_v} \bar{\mathbf{h}}_{uv}, \tag{7}$$

$$\mathbf{m}_{vw}^{\text{edge-embedding-from-edge-states}} = \sum_{u \in \mathcal{N}_v \setminus w} \bar{\mathbf{h}}_{uv}. \tag{8}$$

Then, the edge-view GTransformer transforms the node messages and edge messages through Pointwise Feed Forward layers and Add&LayerNorm to produce the final node embeddings and edge embeddings, respectively.

In summary, the GROVER model outputs four sets of embeddings from two information flows. The node information flow (node GTransformer) maintains node hidden states and finally transform them into another node embeddings and edge embeddings, while the edge information flow (edge GTransformer) maintains edge hidden states and also transforms them into node and edge embeddings. The four sets of embeddings reflect structural information extracted from the two distinct views, and they are flexible to conduct downstream tasks, such as node-level prediction, edge-level prediction and graph-level prediction (via an extra READOUT component).

### A.1 Fine-tuning Model for Molecular Property Prediction

As explained above, given a molecular graph $G_i$ and the corresponding label $\boldsymbol{y}_i$, GROVER produces two node embeddings, $\mathbf{H}_{i,\text{node-view}}$ and $\mathbf{H}_{i,\text{edge-view}}$, from node-view GTransformer and edge-view GTransformer, respectively. We feed these two node embeddings into a shared self-attentive READOUT function to generate the graph-level embedding [54, 28]:

$$\mathbf{S} = \text{softmax}\left(\mathbf{W}_2 \tanh\left(\mathbf{W}_1 \mathbf{H}^\top\right)\right),$$
$$\boldsymbol{g} = \text{Flatten}(\mathbf{SH}), \tag{9}$$

where $\mathbf{W}_1 \in \mathbb{R}^{d_{\text{attn\_hidden}} \times d_{\text{hidden\_size}}}$ and $\mathbf{W}_2 \in \mathbb{R}^{d_{\text{attn\_out}} \times d_{\text{attn\_hidden}}}$ are two weight matrix and $\boldsymbol{g}$ is the final graph embedding. After the READOUT, we employ two distinct MLPs to generate two predictions: $\boldsymbol{p}_{i,\text{node-view}}$ and $\boldsymbol{p}_{i,\text{edge-view}}$. Besides the supervised loss $\mathcal{L}(\boldsymbol{p}_{i,\text{node-view}}, \boldsymbol{y}_i) + \mathcal{L}(\boldsymbol{p}_{i,\text{edge-view}}, \boldsymbol{y}_i)$, the final loss function also includes a disagreement loss [28] $\mathcal{L}_{\text{diss}} = ||\boldsymbol{p}_{i,\text{node-view}} - \boldsymbol{p}_{i,\text{edge-view}}||_2$ to retrain the consensus of two predictions.

### A.2 Constructing Contextual Properties for Edges

In Section 4.2 we describe an example of constructing contextual properties of nodes, here we present an instance of cooking edge contextual properties in order to complete the picture.

Similar to the process of node contextual property construction, we define recurrent statistical properties of local subgraph in a two-step manner. Let us take the graphs in Figure 7 for instance and consider the double chemical bond in red color in the left graph.

Step I: We extract its local subgraph as its $k$-hop neighboring nodes and edges. When $k$=1, it involves the Nitrogen atom, Carbon atom and the two single bonds. Step II: We extract statistical properties

of this subgraph, specifically, we count the number of occurrence of (node, edge) pairs around the center edge, which makes the term of `node-edge-counts`. Then we list all the node-edge counts terms in alphabetical order, which makes the final property: e.g., `DOUBLE_C_SINGLE1_N-SINGLE1` in the example.

Note that there are two graphs and two double bonds in red color in Figure 7, since their subgraphs have the same statistical property, the resulted contextual properties of the two bonds would be the same. For a different point of view, this step can be viewed as a clustering process: the subgraphs are clustered according to the extracted properties, one property corresponds to a cluster of subgraphs with the same statistical property.

## B  Details about Experimental Setup

### B.1  Dataset Description

Table 3: Dataset information

| Type | Category | Dataset | # Tasks | # Compounds | Metric |
|---|---|---|---|---|---|
| Classification | Biophysics | BBBP | 1 | 2039 | ROC-AUC |
| | Physiology | SIDER | 27 | 1427 | ROC-AUC |
| | | ClinTox | 2 | 1478 | ROC-AUC |
| | | BACE | 1 | 1513 | ROC-AUC |
| | | Tox21 | 12 | 7831 | ROC-AUC |
| | | ToxCast | 617 | 8575 | ROC-AUC |
| Regression | Physical chemistry | FreeSolv | 1 | 642 | RMSE |
| | | ESOL | 1 | 1128 | RMSE |
| | | Lipophilicity | 1 | 4200 | RMSE |
| | Quantum mechanics | QM7 | 1 | 6830 | MAE |
| | | QM8 | 12 | 21786 | MAE |

Table 3 summaries information of benchmark datasets, including task type, dataset size, and evaluation metrics. The details of each dataset are listed bellow [60]:

**Molecular Classification Datasets.**

- `BBBP` [32] involves records of whether a compound carries the permeability property of penetrating the blood-brain barrier.

- `SIDER` [26] records marketed drugs along with its adverse drug reactions, also known as the Side Effect Resource .

- `ClinTox` [12] compares drugs approved through FDA and drugs eliminated due to the toxicity during clinical trials.

- `BACE` [49] is collected for recording compounds which could act as the inhibitors of human $\beta$-secretase 1 (BACE-1) in the past few years.

- `Tox21` [1] is a public database measuring the toxicity of compounds, which has been used in the 2014 Tox21 Data Challenge.

- `ToxCast` [41] contains multiple toxicity labels over thousands of compounds by running high-throughput screening tests on thousands of chemicals.

**Molecular Regression Datasets.**

- `QM7` [4] is a subset of GDB-13, which records the computed atomization energies of stable and synthetically accessible organic molecules, such as HOMO/LUMO, atomization energy, etc. It contains various molecular structures such as triple bonds, cycles, amide, epoxy, etc .

- `QM8` [39] contains computer-generated quantum mechanical properties, e.g., electronic spectra and excited state energy of small molecules.

- `ESOL` is a small dataset documenting the solubility of compounds [8].

- `Lipophilicity` [11] is selected from the ChEMBL database, which is an important property that affects the molecular membrane permeability and solubility. The data is obtained via octanol/water distribution coefficient experiments .

- `FreeSolv` [33] is selected from the Free Solvation Database, which contains the hydration free energy of small molecules in water from both experiments and alchemical free energy calculations .

**Dataset Splitting.** We apply the scaffold splitting [2] for all tasks on all datasets. It splits the molecules with distinct two-dimensional structural frameworks into different subsets. It is a more challenging but practical setting since the test molecular can be structurally different from training set. Here we apply the scaffold splitting to construct the train/validation/test sets.

## B.2  Feature Extraction Processes for Molecules

The feature extraction contains two parts: 1) Node / edge feature extraction. We use RDKit to extract the atom and bond features as the input of dyMPN. Table 4 and Tabel 5 show the atom and bond feature we used in GROVER. 2) Molecule-level feature extraction. Following the same protocol of [63, 60], we extract additional 200 molecule-level features by RDKit for each molecule and concatenate these features to the output of self-attentive READOUT, to go through MLP for the final prediction.

Table 4: Atom features.

| features | size | description |
|---|---|---|
| atom type | 100 | type of atom (e.g., C, N, O), by atomic number |
| formal charge | 5 | integer electronic charge assigned to atom |
| number of bonds | 6 | number of bonds the atom is involved in |
| chirality | 5 | number of bonded hydrogen atoms |
| number of H | 5 | number of bonded hydrogen atoms |
| atomic mass | 1 | mass of the atom, divided by 100 |
| aromaticity | 1 | whether this atom is part of an aromatic system |
| hybridization | 5 | sp, sp2, sp3, sp3d, or sp3d2 |

Table 5: Bond features.

| features | size | description |
|---|---|---|
| bond type | 4 | single, double, triple, or aromatic |
| stereo | 6 | none, any, E/Z or cis/trans |
| in ring | 1 | whether the bond is part of a ring |
| conjugated | 1 | whether the bond is conjugated |

## C  Implementation and Pre-training Details

We use Pytorch to implement GROVER and horovod [47] for the distributed training. We use the Adam optimizer with learning rate $0.00015$ and L2 weight decay for $10^{-7}$. We train the model for 500 epochs. The learning rate warmup over the first two epochs and decreases exponentially from $0.00015$ to $0.00001$. We use PReLU [16] as the activation function and the dropout rate is 0.1 for all layers. Both GROVER$_{base}$ and GROVER$_{large}$ contain 4 heads. We set the iteration $L = 1$ and sample $K_l \sim \phi(\mu = 6, \sigma = 1, a = 3, b = 9)$ for the embedded dyMPN in GROVER. $\phi(\mu, \sigma, a, b)$ is a truncated normal distribution with a truncation range $(a, b)$. The hidden size for GROVER$_{base}$ and GROVER$_{base}$ are 800 and 1200 respectively.

## D    Fine-tuning Details

For each task, we try 300 different hyper-parameter combinations via random search to find the best results. Table 6 demonstrates all the hyper-parameters of fine-tuning model. All fine-tuning tasks are run on a single P40 GPU.

Table 6: The fine-tuning hyper-parameters

| hyper-parameter | Description | Range |
|---|---|---|
| batch_size | the input batch_size. | 32 |
| init_lr | initial learning rate ratio of Noam learning rate scheduler. The real initial learning rate is max_lr / init_lr. | 10 |
| max_lr | maximum learning rate of Noam learning rate scheduler. | $0.0001 \sim 0.001$ |
| final_lr | final learning rate ratio of Noam learning rate scheduler. The real final learning rate is max_lr / final_lr. | $2 \sim 10$ |
| dropout | dropout ratio. | 0, 0.05, 0.1,0.2 |
| attn_hidden | hidden size for the self-attentive readout. | 128 |
| attn_out | the number of output heads for the self-attentive readout. | 4,8 |
| dist_coff | coefficient of the disagreement loss | 0.05, 0.1,0.15 |
| bond_drop_rate | drop edge ratio [43] | 0, 0.2,0.4,0.6 |
| ffn_num_layer | The number of MLP layers. | 2,3 |
| ffn_hidden_size | The hidden size of MLP layers. | 5,7,13 |

## E    Additional Experimental Results

### E.1    Effect of Self-supervised Pre-training on Regression Tasks

Table 7 depicts the additional results of the comparison of the performance of pre-trained GROVER and GROVER without pre-training on regression tasks.

Table 7: Comparison between GROVER with and without pre-training on regression tasks

|  |  | GROVER | No Pre-training | Absolute Improvement |
|---|---|---|---|---|
| RMSE | FreeSolv | **1.544** | 1.987 | 0.443 |
|  | ESOL | **0.831** | 0.911 | 0.080 |
|  | Lipo | **0.560** | 0.643 | 0.083 |
| MAE | QM7 | **72.600** | 89.408 | 16.808 |
|  | QM8 | **0.013** | 0.017 | 0.004 |

### E.2    GROVER **Fine-tuning Tasks with Other Backbones**

In order to verify the effectiveness of the proposed self-supervised tasks, we report the fine-tuning results by Hu et al. with and without pre-training in Table 8. As a comparison, we also involve the performance of GROVER with the backbone GIN and MPNN trained in Section 5.2. We find that without pre-training, our GROVER-GIN is consistent with Hu et al. on average, thus verifying the reliability of our implementations. However, after pre-training, GROVER-GIN achieves nearly 2% higher number than Hu et al., which supports the advantage of our proposed self-supervised loss.

Table 8: Comparison between different methods. The metric is AUC-ROC. The numbers in brackets are the standard deviation.

|  | Hu. et al. | | GROVER-GIN | | GROVER-MPNN | |
|---|---|---|---|---|---|---|
|  | w pre-train | w/o pre-train | w pre-train | w/o pre-train | w pre-train | w/o pre-train |
| BBBP | $0.915_{(0.040)}$ | $0.899_{(0.035)}$ | $0.925_{(0.036)}$ | $0.901_{(0.051)}$ | $0.929_{(0.029)}$ | $0.917_{(0.027)}$ |
| SIDER | $0.614_{(0.006)}$ | $0.615_{(0.007)}$ | $0.648_{(0.015)}$ | $0.627_{(0.016)}$ | $0.650_{(0.003)}$ | $0.637_{(0.030)}$ |
| BACE | $0.851_{(0.027)}$ | $0.837_{(0.028)}$ | $0.862_{(0.020)}$ | $0.823_{(0.050)}$ | $0.872_{(0.031)}$ | $0.852_{(0.034)}$ |
| Average | 0.793 | 0.784 | 0.812 | 0.784 | 0.817 | 0.802 |