[Reviews · NeurIPS 2020]

Review 1

Summary and Contributions: The paper presents a new pre-training strategy and dataset for Graph NNs, as well as a new GNN/transformer architecture. The GTransformer alternates K-hop aggregation (with randomized K) over the graph with dense attention over all nodes in every layer. The new techniques are shown to have good performance gains across the MoleculeNet benchmarks. The paper builds on, and closely follows, Hu et al ICLR 2020 "strategies for pre-training graph neural networks".

Strengths: * The GTransformer is an interesting new architecture. I believe the alternation between graph-structured / global aggregation is a promising idea. * The proposed self-supervision tasks is an interesting contribution and convincingly (as per Table2) contributes to the success on the benchmarks * Results seem strong, beating all baselines across the panel (caveat: I'm not too familiar with the benchmark dataset, and see question in "Correctness" about the numbers)

Weaknesses: See some questions about correctness, and missing baselines below. There is a lot of room for improvement when it comes to the writing; * Mainly in terms of clarity (see below) - the model architecture is very hard to understand. * writing seems very rushed, with unfinished sentences (eg L332), * strange turns of phrases (eg "largest dataset that we have ever met", "largely-expressive models", "dark clouds") * There are sometimes long-winded introductory sentences without real information, which gets in the way of clear exposition (eg L171-173). I encourage the authors to rewrite the paper in a more direct style, removing all unnecessary adjectives.

Correctness: * Table 1: which metric is reported? I cross-checked with the numbers in Hu et al ICLR 2020 "strategies for pre-training graph neural networks", and the numbers reported here for that baseline don't match (eg BBBP 0.915 vs 68.7 in Hu et al). Of course the metric should be explicitly written in table caption and such strange discrepancies addressed in the text. * The ablations in Table 2 and Figure 4 are good, however I still miss a more standard graph neural network architecture like GAT or GIN pre-trained on the same self-supervised task. Likewise, a column should be added to Table 2 to compare with pre-training following the strategy of Hu et al 2020

Clarity: Even though the paper shows many promising results and has interesting novel ideas, the general writing style is poor and the paper needs thorough rewriting. See also sone notes under "weaknesses" The introduction of GTransformer is not comprehensible L161-170. * a number of high-level ideas are mentioned which make no sense before reading the details. eg "dyMPN" is mentioned without short summary, then referenced several more times before explaining * Comparing the "transformer layer" (multi-head attention layer?) to GAT is somewhat misleading - the crucial contribution of GAT is to *restrict* attention to the graph structure. It is not clearly explained in which phases of the architecture information flow is restricted by the graph connectivity structure, and which parts go over every node/edge in the graph. It is somewhat described on L174 onwards (bi-level), but this should be front and center in Sec4.1. IMO the first expectation of readers would be that the transformer attention would be restricted over the graph structure as in GAT, so clearly state that is not the case. Let me try to rephrase the outline of the architecture: > "The core innovation of the GTransformer is to alternate K-hop aggregation (with randomized K) over the graph with dense attention over all nodes. A single layer (name this as "GTransformer layer"?) consists of: > * dyMPN: message passing via graph edges with a randomized number of hops. > * Attention: Q K V per node, with dense self-attention (all nodes attend over *all* nodes, ignores graph structure) > * 1-hop "Aggregate2Node" Some crucial details I could not find back: * for dyMPN, which aggregation strategy is used? Same question for aggregate2node, aggregate2edge. * L323: which value for Kl is chosen? * How many "GTransformer layers" as depicted in Figure 1? Another unclarity about the architecture: I wonder whether the node and edge GTransformer components have any state exchange or parameter sharing between them? If yes, where and how (between every GTransformer layer)? If no; how is the state combined for the final task at hand? How is performance impacted if only the node GTransformer is used? GNN prelims: L139; worth to mention that hops within an iteration (later on: iteration is renamed layer) use the same weights, while between iterations/layers weights are different. Appendix L21, edge-embed-from-node-states; for edge m_{vw} only a sum over N_{v} \ w is done, one could point explicitly that this is asymmetric in v,w, and that *both* edge directions have separate hidden states.

Relation to Prior Work: I see no issues. One pointer: Ingraham et al (NeurIPS 2019) uses an auto-regressive transformer where attention is restricted to the graph; in a protein generation context.

Reproducibility: No

Additional Feedback: I recommend weak reject because of severe issues with the paper writing and clarity, however with proper presentation, I find the contributions of the paper would be strong enough to recommend accept. Some more minor pointers where writing can be improved: * Table 1 caption is bad. The caption should stand on its own, eg: "Performance metrics on Moleculenet. On the classification task we report ROC-AUC, on the regression task MSE(?). Summarize observations and significance" * STOA for state of the art -- should be SOTA * L234: "Suppose _we_ randomly choose" The acronym to name mapping is far-fetched. I'd recommend finding a better finding acronym with a fitting name. -- UPDATE after author feedback -- The author rebuttal new results are quite good, it addresses the major coupling concern myself + R2 had, and shows decoupled gains from both architecture and pre-training strategy. * pre-training gives a gain on different GNN architectures, and * GTransformer does better on pretraining objective than GIN or MPNN The response also addressed my concern about correctness / discrepancy due to different test split. My other main concern about writing and clarity can of course not properly be addressed in the one-page author response, but I am counting on the thorough rewrite promised in the rebuttal, and increase my score from 5 to 6. --- end update ---


Review 2

Summary and Contributions: This paper proposes a BERT-style pre-training method for graph neural networks in the context of molecular property prediction. The proposed method demonstrates clear improvement on MoleculeNet benchmark compared to various state-of-the-art baselines.

Strengths: == Soundness of claims == The empirical evaluation is extensive and comprehensive. It incorporates most of the state-of-the-art baselines that needs to be compared. The evaluation is based on scaffold splitting and follows the standard evaluation protocol in property prediction field. The results are convincing enough to show that pre-training works for property prediction. == Relevance to NeurIPS == AI-based drug discovery is an important application (or emerging field) in machine learning community. Pre-training for molecular property prediction is an important problem for drug discovery because most property prediction datasets are rather small (1k-10k datapoints). I believe this is very relevant to NeurIPS.

Weaknesses: == Novelty of Contribution == The method itself is not novel. I think atom / subgraph masking has already been proposed in Hu et al., 2020. The pre-training strategy is conceptually the same as Hu et al., 2020, albeit with some minor differences. The GNN architecture is different from Hu et al. The dynamic MPN and the transformer layer on top of dyMPN seems to be new, but this is not the focus of this paper. == Does pre-training only work for dyMPN? == The pre-training strategy works well under this specific architecture. However, it is unclear whether this works for general GNN architectures. It seems like this pre-training only works under dyMPN + Transformer. Without dyMPN + Transformer, does pre-training still outperforms state-of-the-art MPNNs such as D-MPNN or attentionFP? The paper does not address this important question. == Reproducibility == Seems like the code is not attached in the submission. I hope that the experiments / code can be open-sourced later.

Correctness: The major claim of the paper and its empirical methodology is correct. However, there are some claims in the paper that lacks justification. For example, in line 189, authors claim that their long-range residual connection can alleviate the over-smoothing. But there is no justification for that.

Clarity: The paper is clear but some details are hidden or deferred to the appendix. For example, section 4.3 seems pretty vague. Eq.(3) only discuss how node embeddings are pooled. How about edge embeddings (as shown in Figure 1)?

Relation to Prior Work: Yes, the related work section clearly discusses the difference from previous works. I think the related work section covers enough prior works related to molecule property prediction to my knowledge.

Reproducibility: No

Additional Feedback: 1. Is there any experiments justifying the architectural choices? Such as long-range residual connection versus multiple short-range connections? 2. In Figure 1, in the output layer, I see both node / edge embeddings. How are these embeddings aggregated into graph-level representation? Eq.(3) only mentions the node embedding pooling operation. === Post Rebuttal === I have read author's rebuttal and I will keep my original score.


Review 3

Summary and Contributions: This work introduces a GNN based on transformers' paradigm for pre-training and fine-tuning on molecular classification and regression tasks.

Strengths: This is an interesting work that applies recent successful ideas in NLP from Transformers/BERT to molecular tasks, and shows it works well on several molecular benchmarks.

Weaknesses: No strong weakness. Ideas are borrowed from NLP and seems well executed in the context of molecular graphs. Performances are good although the standard deviation can be large for some datasets on Table 1. Baseline techniques are not well discussed. The number of parameters for each technique must be given to compare fairly with the 48M and 100M of GROVER. Also, are the hyper-parameters of the baseline techniques optimized for fair comparisons? The average molecule size for each dataset should be reported. The (large) computational time should also be reported in the paper, not in the supplementary: "We use 250 Nvidia V100 GPUs to pre-train GROVERbase and GROVERlarge. Pre-training 116 GROVERbase and GROVERlarge took 2.5 days and 4 days respectively." A discussion on reinforcement learning for molecular tasks would be good s.a. Jiaxuan You, Bowen Liu, Zhitao Ying, Vijay Pande, and Jure Leskovec. Graph convolutional policy network for goal-directed molecular graph generation. In Advances in Neural Information Processing Systems, pages 6412–6422, 2018.

Correctness: Likely.

Clarity: Some parts are not written formally, and should be improved s.a. "dark coulds" or "what's worse".

Relation to Prior Work: Seems fine but some references are missing: - Nicola De Cao and Thomas Kipf. Molgan: An implicit generative model for small molecular graphs. arXiv preprint arXiv:1805.11973, 2018. - Wengong Jin, Regina Barzilay, and Tommi Jaakkola. Junction tree variational autoencoder for molecular graph generation. arXiv preprint arXiv:1802.04364, 2018. - Matt J Kusner, Brooks Paige, and José Miguel Hernández-Lobato. Grammar variational autoencoder. In Proceedings of the 34th International Conference on Machine Learning-Volume 70, pages 1945–1954. JMLR, 2017.

Reproducibility: No

Additional Feedback: ----- Update after author response ----- I want to thank the authors for their rebuttal. However, I have doubts about the efforts spent to improve the baseline techniques and compare fairy with the proposed model. For this reason, I decided to downgrade my score. ----- Update after author response -----


Review 4

Summary and Contributions: This paper proposed a pre-training Graph Representation from self-supervised message passing transformer framework (GROVER) for molecular representation. GROVER integrates a message passing networks with the Transformer architecture(node and edge). Structural and semantic information can be learned from a large unlabelled molecular data by self-supervised learning.

Strengths: The model is a Transformer-style networks with Graph neural networks as self-attention building blocks. The pre-training architecture with self-supervision, message passing networks and GNN transformer are clearly described in detail. 6% improvement is achieved on average compared with other STOA methods.

Weaknesses: The architectural difference with respect to the graph attention network shall be discussed. The authors responded to this in the response and promised to add this in the final version.

Correctness: The claims and method are correct.

Clarity: The paper is well written.

Relation to Prior Work: The related work section is comprehensive.

Reproducibility: Yes

Additional Feedback:

[Author Response · NeurIPS 2020]

Table A: Comparison between different methods. The metric is AUC-ROC. The numbers in brackets are the standard deviation.

| | Hu. et al. | | GROVER-GIN | | GROVER-MPNN | |
|---|---|---|---|---|---|---|
| | w pre-train | w/o pre-train | w pre-train | w/o pre-train | w pre-train | w/o pre-train |
| BBBP | $0.915_{(0.040)}$ | $0.899_{(0.035)}$ | $0.925_{(0.036)}$ | $0.901_{(0.051)}$ | $0.929_{(0.029)}$ | $0.917_{(0.027)}$ |
| SIDER | $0.614_{(0.006)}$ | $0.615_{(0.007)}$ | $0.648_{(0.015)}$ | $0.627_{(0.016)}$ | $0.650_{(0.003)}$ | $0.637_{(0.030)}$ |
| BACE | $0.851_{(0.027)}$ | $0.837_{(0.028)}$ | $0.862_{(0.020)}$ | $0.823_{(0.050)}$ | $0.872_{(0.031)}$ | $0.852_{(0.034)}$ |
| Average | 0.793 | 0.784 | 0.812 | 0.784 | 0.817 | 0.802 |

Figure A: The training and validation losses on different architectures.

We sincerely thank all reviewers for their valuable comments. Below are our point-by-point responses.

**To Reviewer #1:** Thank you for your detailed and valuable suggestions in terms of our writing. In our final version, we will carefully rephrase the introduction of GTransformer (*e.g.* explaining more on high-level concepts, fixing the comparisons with GAT, and rephrasing the outline of the architecture), rewrite and polish the sentences in Section 3 and 4, and remove all unnecessary adjectives to make our paper more direct and concise. We believe such revisions are achievable and do not change the main story of our paper. We address the major concerns as follows. **Q1. The metric in Table 1.** We use ROC-AUC and follow the benchmarking n-fold scaffold setting for train/test split, while Hu et al. report ROC-AUC (%) under a different split strategy. Thus, it's reasonable to produce different numbers for the same method. We will specify this in the caption of Table 1. **Q2. More ablations. 1.** We implement GIN and MPNN based on our framework, and per-train them under the same setting as in Section 5.2. For a fair comparison, we set a large hidden size to ensure them to be nearly as big as GROVER (38M parameters in Section 5.2). As shown in Fig. A, our model still outperforms GIN and MPNN in both training and validation, which again verifies the effectiveness of our GTransformer. **2.** We report the results by Hu et al. with and without pre-training in Tab. A. As a comparison, we also involve the performance of GROVER with the backbone GIN. We find that without pre-training, our GROVER-GIN is consistent with Hu et al. on average, thus verifying the reliability of our implementations. However, after pre-training, GROVER-GIN achieves nearly 2% higher number than Hu et al., which supports the advantage of our proposed self-supervised loss. **Q3. On the node and edge GTransformers.** Sorry for the confusion. There is no state exchanging or parameter sharing between these two GTransformers. We apply the disagreement loss on the predictions of the two views to retain the consensus (See L46 in the supplement). We will clarify this point in the final version. **Q5. Other details.** We use SUM aggregation for dyMPN, aggregate2node, and aggregate2edge. In L323, $K_l = 6$ is the mean of the sample distribution in dyMPN (See L112 in the supplement). Fig. 1 depicts only one GTransformer layer. The prior work (Ingraham et al) will be added. We will revise our paper to reflect the above details in the final version.

**To Reviewer #2:** **Q1. Novelty of contribution.** We would like to highlight that our method clearly differs from the paper by Hu et al. in three aspects: 1. Hu et al. mask only nodes/edges to predict their types, while our subgraph masking task predicts the types of nodes, edges and local structures, which enables to encode more information. 2. Regarding graph-level pre-training, our method is completely unsupervised, whereas Hu et al. implement the full-supervised strategy in the experiments, which would limit the practical usage. 3. The development of GTransformer is novel and crucial. Considering the extra ablations in Fig. A, Our GTransformer achieves better performance than GIN and MPNN provided the same pre-training scenario. Overall, we believe our contributions are novel and remarkable. **Q2. Does pre-training only work for** dyMPN? Not really, and it works for general GNNs. To show this, we have conducted MPNN and GIN with and without pre-training in Tab. A following the same protocol in Section 5.2. The results read that the pre-training stage is also crucial for these two models. **Q3. Justification on alleviating over-smoothing.** The claim that long-range residual connections will alleviate over-smoothing is directly borrowed from JKNet [Xu et al. ICML-18] which will be cited. **Q4. Justification of architectural choices.** Indeed, GROVER-w/o-GTrans represents the traditional transformer with multiple short-range connections. In Section 5.2, we have conducted a comparison between GROVER and GROVER-w/o-GTrans, which can be regarded as a long-range residual connection versus multiple short-range connections. We will further specify this in the final version. **Q5. Other issues.** For the pre-training task, we use MEAN readout function to generate the graph-level representation. We will release the code as well as the pre-train models to facilitate the follow-up researches.

**To Reviewer #3:** **Q1. Baselines and hyper-parameters.** For the baseline methods, we first follow the suggested hyper-parameters given by the original implementation, and if the results cannot be reproduced, we will conduct hyper-parameter tuning by validation. We will clarify this and provide the parameter-size of each baseline in Table 1 of the final version. Most baselines have smaller architecture than ours, and even we simply enlarge their model sizes, the performances are still worse than our method (see our response to Q2, R#1 and the extra ablations in Fig. A). **Q2. Other issues.** In the final version, we will report the average molecule size for each dataset, move the computational cost of the pre-training models from supplementary to the paper, cite and discuss the raised references by the reviewer.

**To Reviewer #4:** Thanks for the comments. In detail (as also supported by R1), GAT restricts the attention to each existing edge, while in our model, the transformer-style layer acts as a dense attention over all nodes to capture the graph-level information. We are willing to discuss the difference between our method and GAT in the final version.

[Meta-Review · NeurIPS 2020]

This paper presents a few ideas for self-supervised pre-training of GNNs as well as a new GNN + transformer model, the results look solid when compared against other baselines. The reviewers have a lukewarm reaction to the paper but unanimously support an accept. I hope the authors can improve the clarity of the paper and include the additional results obtained during rebuttal into the paper in the final version.